# Vitamin D and Colorectal Cancer Prevention: Immunological Mechanisms, Inflammatory Pathways, and Nutritional Implications

**DOI:** 10.3390/nu17081351

**Published:** 2025-04-15

**Authors:** Mónika Fekete, Andrea Lehoczki, Ágnes Szappanos, Virág Zábó, Csilla Kaposvári, Alpár Horváth, Árpád Farkas, Vince Fazekas-Pongor, Dávid Major, Ágnes Lipécz, Tamás Csípő, János Tamás Varga

**Affiliations:** 1Institute of Preventive Medicine and Public Health, Faculty of Medicine, Semmelweis University, 1089 Budapest, Hungary; fekete.monika@semmelweis.hu (M.F.); ceglediandi@freemail.hu (A.L.); kaposvari.csilla@semmelweis.hu (C.K.); pongor.vince@semmelweis.hu (V.F.-P.); major.david@semmelweis.hu (D.M.); lipecz.agnes@semmelweis.hu (Á.L.); csipo.tamas@semmelweis.hu (T.C.); 2Health Sciences Division, Doctoral College, Semmelweis University, 1085 Budapest, Hungary; zabo.virag@semmelweis.hu; 3Heart and Vascular Center, Semmelweis University, 1122 Budapest, Hungary; drszappanos@gmail.com; 4Department of Rheumatology and Clinical Immunology, Semmelweis University, 1023 Budapest, Hungary; 5Pulmonology Center of the Reformed Church in Hungary, 2045 Törökbálint, Hungary; horvath65881@gmail.com; 6HUN-REN Centre for Energy Research, 1121 Budapest, Hungary; farkasar@gmail.com; 7Department of Pulmonology, Semmelweis University, 1083 Budapest, Hungary

**Keywords:** vitamin D, colorectal cancer, immunomodulation, inflammation, carcinogenesis, nutrition, CRC prevention

## Abstract

Vitamin D plays a crucial role in the regulation of the immune system, with immunomodulatory effects that are key in the prevention of colorectal cancer (CRC). Over the past decades, research has shown that this steroid hormone impacts much more than bone health, significantly influencing immune responses. Vitamin D enhances immune organ functions such as the spleen and lymph nodes, and boosts T-cell activity, which is essential in defending the body against tumors. Additionally, vitamin D mitigates inflammatory responses closely linked to cancer development, reducing the inflammation that contributes to CRC. It acts via vitamin D receptors (VDRs) expressed on immune cells, modulating immune responses. Adequate vitamin D levels influence gene expression related to inflammation and cell proliferation, inhibiting tumor development. Vitamin D also activates mechanisms that suppress cancer cell survival, proliferation, migration, and metastasis. Low levels of vitamin D have been associated with an increased risk of CRC, with deficiency correlating with higher disease incidence. Lifestyle factors, such as a diet high in red meat and calories but low in fiber, fruits, and vegetables, as well as physical inactivity, contribute significantly to CRC risk. Insufficient calcium and vitamin D intake are also linked to disease occurrence and poorer clinical outcomes. Maintaining optimal vitamin D levels and adequate dietary intake is crucial in preventing CRC and improving patient prognosis. This review explores the role of vitamin D in immune regulation and summarizes findings from randomized clinical trials assessing the effects of vitamin D supplementation on CRC outcomes.

## 1. Introduction

The biological effects of vitamin D, particularly its anti-cancer properties, have been the focus of extensive research over the past few decades [1,2,3]. Unlike traditional vitamins, vitamin D can be synthesized endogenously when the skin is exposed to sufficient UV-B radiation [4]. This steroid hormone precursor is derived from 7-dehydrocholesterol in the skin under sunlight exposure [5]. The primary form, vitamin D_3_ (cholecalciferol), plays a crucial role in calcium and phosphorus homeostasis while also participating in key physiological processes, including immunomodulation, the reduction in inflammation, induction of apoptosis, and anti-angiogenic effects [6,7].

There are two main forms of vitamin D: D_3_, which is found in animal-derived foods, and D_2_, which is produced in plants and fungi [8]. Although D_2_ contributes to vitamin D status, it is significantly less effective than D_3_, making the latter the preferred form [9]. The efficiency of cutaneous vitamin D synthesis depends on sunlight exposure, but factors such as skin pigmentation, sunscreen use, and age can influence this process. During the winter months, especially at latitudes above the 35th parallel, sunlight exposure is insufficient for adequate synthesis, necessitating supplementation [4].

The biologically active form of vitamin D, 1,25-dihydroxyvitamin D (1,25(OH)_2_D), is synthesized in the kidney under strict hormonal regulation [10]. The process of 1α-hydroxylation in the kidney is modulated by tissue-specific factors, including inflammatory cytokines, which enable local autocrine and paracrine functions of vitamin D [11]. These effects influence cell division, differentiation, and apoptosis. Vitamin D exerts both genomic and non-genomic effects [12]. The genomic effects are mediated by vitamin D receptors (VDRs), leading to transcriptional responses that occur over a longer period and primarily influence gene expression. This allows vitamin D to play a crucial role in regulating various physiological processes, such as calcium metabolism and immune responses. In contrast, the non-genomic effects are faster and generate immediate responses, as the VDRs are located on the cell membrane, and their activation sends quick signals to the cell. These pathways play a key role in vitamin D’s rapid responses, such as calcium mobilization and managing oxidative stress [13]. These two mechanisms of action allow vitamin D to have both long-term and short-term effects, playing a critical role in maintaining healthy cellular functions, and in the prevention and treatment of various diseases.

Vitamin D deficiency is a global health concern and has been linked to various chronic diseases, including malignancies [14,15]. The association between vitamin D and colorectal cancer (CRC) is well documented in numerous epidemiological and preclinical studies [16,17]. Through its antioxidant, anti-inflammatory, and anti-carcinogenic properties, vitamin D inhibits cancer cell proliferation, promotes differentiation, and reduces tumor invasiveness and metastatic potential [18].

This review aims to investigate the physiological functions and anti-cancer effects of vitamin D, with a particular focus on its role in CRC. In addition, we explore the prevalence of vitamin D deficiency and summarize clinical trials evaluating the impact of vitamin D supplementation on CRC outcomes, providing an analysis of the latest research findings.

## 2. Methods

A comprehensive literature search was conducted in the PubMed, Web of Science and Scopus databases, covering studies published from inception until 1 January 2025. The primary objective of this research was to investigate the role of vitamin D in colorectal cancer prevention, with a particular focus on its risk-reducing effects, immunological mechanisms, and anti-inflammatory properties.

The search strategy utilized predefined Medical Subject Headings (MeSH) terms and keywords, including vitamin D, colorectal cancer, colorectal adenomas, colorectal polyps, risk, immunomodulation, inflammation, carcinogenesis, CRC prevention, dietary intake of vitamin D, and vitamin D supplementation, combined with AND and OR operators. Duplicate records were removed, and after screening titles and abstracts, we excluded studies that did not meet inclusion criteria. The remaining articles underwent full-text evaluation.

The literature review followed the PICO (Population, Intervention, Comparison, Outcome) framework (Table 1). Relevant studies were selected based on the following criteria:

In addition, findings from randomized controlled trials and human clinical studies were summarized in tables. The results were presented using hazard ratios (HRs), odds ratios (ORs), and confidence intervals (CIs), which were analyzed narratively. This review article includes a total of 50 clinically relevant cohort studies, organized in tables, comprising 1,305,997 participants. The study selection process is illustrated in Figure 1.

## 3. Prevalence of Vitamin D Deficiency

Vitamin D deficiency is one of the most widespread nutritional and health concerns today, posing a significant global public health challenge [19]. It is estimated that a large proportion of the world’s population is affected, particularly in temperate regions where sunlight exposure is limited, as well as in areas where high skin pigmentation or cultural practices restrict sun exposure [14]. On a global scale, the prevalence of vitamin D deficiency is estimated to range from 30% to 50%, with even higher rates seen in specific populations [20].

Several factors contribute to vitamin D deficiency, including gender, seasonality, age, and economic status [19,21]. The condition is more prevalent in women, partly due to conservative clothing practices. Additionally, vitamin D deficiency is more common during the winter months, when sunlight exposure is reduced. It is also particularly common among adults aged 19 to 44 years, likely due to a combination of increased muscle mass and inadequate vitamin D intake. Furthermore, vitamin D deficiency is more frequently observed in low- and middle-income countries, where access to adequate nutrition and supplements may be limited.

Certain populations are particularly vulnerable to vitamin D deficiency. Urban populations, where air pollution and dense infrastructure limit sunlight exposure, are at greater risk. The elderly are also more susceptible, as their ability to synthesize vitamin D through skin exposure significantly declines with age. In regions such as Asia and the Middle East, a combination of high skin pigmentation and conservative clothing practices further contributes to lower vitamin D levels.

In Europe, vitamin D deficiency remains a significant concern. Data suggest that 18% of the population has serum vitamin D levels below 30 nmol/L, while 53% have levels below 50 nmol/L. Similar deficiencies have been reported in Southeast Asia (22%) and the Western Pacific region (10%) [22]. Approximately 40% of Europeans are considered vitamin D deficient, with 13% classified as severely deficient.

In Hungary, the prevalence of vitamin D deficiency is even more pronounced, particularly during the winter months. Studies indicate that by early spring, over 70% of the Hungarian population is affected, highlighting the critical importance of seasonal vitamin D supplementation [23,24].

## 4. Prevalence of Colorectal Cancer

CRC is one of the most pressing global public health concerns, particularly in countries that have adopted a Western lifestyle and diet, where its incidence has been steadily rising [25]. Each year, approximately 1.2 million new CRC cases are diagnosed worldwide, making it the second most common cancer in men and the third most common in women [23,24].

Over the past decades, the global incidence of CRC has increased significantly, largely due to the rising case numbers in developing countries. The growing adoption of Western dietary and lifestyle patterns in these regions has played a major role in this trend. The mortality rate is equally concerning, with a global CRC mortality rate of 7 per 100,000 population, placing a substantial burden on healthcare systems [26,27].

In Europe, CRC remains a major health issue, ranking among the most prevalent cancers on the continent [28]. The incidence is particularly high in Central Europe, where unfavorable mortality rates further exacerbate the public health challenge [28]. Hungary stands out among European countries, ranking first in incidence and second in age-standardized mortality. The country’s incidence rate is 45.3 per 100,000 population, while the mortality rate reaches 20.2 per 100,000—both figures significantly exceeding the international averages of 19.5 per 100,000 and 7 per 100,000, respectively. In Hungary, the annual CRC mortality rate ranges between 4 and 10 per 10,000 population, markedly higher than the European average [26,27].

Several environmental and genetic factors contribute to CRC development. First, CRC is considered an age-related disease [29,30,31] and its pathogenesis involves cellular and molecular mechanisms of aging [32,33,34]. Environmental [35] and lifestyle factors that accelerate biological mechanisms of aging [36,37,38,39,40,41], play a major role in the pathogenesis of CRC [42,43,44]. For example, a sedentary lifestyle, along with a diet high in fats, red meat, and low in fiber [45,46], significantly contributes to accelerated aging and CRC development. There is also genetic susceptibility, individuals with a family history of CRC have a higher predisposition to the disease. Changes in the microbiome [47,48,49], including increased prevalence of oncobacteria also play a pathogenic role [50,51]. Comorbidities such as inflammatory bowel diseases, obesity [52,53,54], frailty [55], and diabetes [56] also elevate CRC risk.

## 5. The Biological Effects of Vitamin D

The active form of vitamin D, calcitriol, binds to the VDR, a steroid hormone receptor that regulates gene expression [11]. While the VDR is most abundant in the intestines, it plays a regulatory role in various organ systems [57,58,59,60,61]. Calcitriol exerts a range of biological effects, influencing crucial physiological processes. It regulates calcium and phosphorus metabolism by controlling the intestinal absorption of these minerals, mobilizing calcium from bones, and facilitating renal reabsorption [62]. These actions help maintain proper serum and extracellular calcium and phosphorus concentrations, essential for bone metabolism and other metabolic functions [63,64].

Additionally, calcitriol plays a key role in immunomodulation by reducing inflammation, suppressing autoimmune responses, and supporting the function of immune cells [65]. It also impacts cell proliferation and differentiation, inhibiting uncontrolled cell growth and promoting the differentiation of both normal and cancerous cells [66]. Since many tumor cells express VDRs, calcitriol can directly exert anti-tumor effects [67]. Furthermore, it inhibits angiogenesis—the formation of new blood vessels that support tumor growth—and promotes apoptosis, or programmed cell death, both critical in tumor suppression [68].

Recent research has shown that the physiological roles of calcitriol extend beyond calcium and phosphorus homeostasis, influencing significant immunoregulatory, anti-aging, and anti-cancer effects [69,70,71,72,73]. Vitamin D is crucial for maintaining overall health, particularly in bone metabolism, immune function, and cancer prevention [74,75]. However, its effects go well beyond these functions, impacting several other vital systems and physiological processes.

In the nervous system, vitamin D plays an essential role in supporting communication between nerve cells [71,76], helping to maintain mental health and potentially protecting against neurodegenerative diseases such as Alzheimer’s [77]. Due to its impact on the brain, vitamin D has been linked to the preservation of cognitive functions and may reduce the risk of depression as well [78].

In cardiovascular health, vitamin D’s role is also significant [79,80]. It contributes to blood pressure regulation, reduces the risk of hypertension, and helps prevent cardiovascular diseases like heart attacks and arteriosclerosis [81]. Furthermore, it influences lipid profiles, improving cholesterol levels in the blood and lowering the likelihood of heart diseases [82].

In the respiratory system, vitamin D helps prevent respiratory infections such as colds, the flu, and COVID-19 by enhancing immune responses and improving the protection of respiratory epithelial cells [83,84,85,86]. Some studies have also shown that vitamin D may be beneficial in managing chronic respiratory diseases like asthma by reducing inflammation and promoting the relaxation of airway smooth muscles [87,88,89,90,91,92].

In the digestive system, vitamin D plays a crucial role in maintaining gut health. It supports the balance of the gut microbiome, which influences nutrient absorption and helps prevent intestinal inflammation [93,94]. Additionally, vitamin D may reduce the risk of inflammatory bowel diseases such as Crohn’s disease and ulcerative colitis [95].

The thyroid system is also affected by vitamin D, as it helps maintain hormonal balance. Vitamin D contributes to the proper levels of thyroid hormones, which are essential for metabolism [96].

At the metabolic level, vitamin D plays a role in regulating blood sugar levels, thereby contributing to the prevention and management of diabetes [97]. It improves insulin sensitivity and helps regulate glucose metabolism, which can reduce the risk of developing type 2 diabetes [98].

Furthermore, vitamin D has anti-aging effects [99]. As we age, cellular functions slow down, but vitamin D, through its antioxidant and anti-inflammatory properties, can mitigate the harmful effects of aging, helping to maintain overall health. Its anti-aging role helps prevent age-related diseases like osteoporosis or muscle weakness [100,101,102].

Overall, vitamin D plays a fundamental role in the function of numerous systems and physiological processes. The broad range of effects highlights the importance of vitamin D in maintaining health and underscores its protective role in preventing chronic diseases, including cancer.

## 6. The Role of Vitamin D in Colorectal Cancer Prevention

Research indicates that maintaining adequate vitamin D levels may reduce CRC risk. A meta-analysis showed that individuals with higher serum 25(OH)D levels had a 39% lower risk of CRC in case–control studies and a 20% reduced risk in prospective cohort studies [103]. This association was first noted by Garland et al., who found higher CRC mortality rates in regions with lower sunlight exposure and vitamin D levels [104]. Subsequent studies have consistently validated the protective role of vitamin D in CRC.

Preclinical studies have demonstrated that calcitriol, the active form of vitamin D, exerts anti-tumor effects through several CRC-specific mechanisms, including the inhibition of cancer cell proliferation, induction of apoptosis, promotion of cellular differentiation, and antiangiogenesis. Calcitriol also regulates genes involved in cell cycle control, apoptosis, and immune responses while influencing the tumor microenvironment through autocrine and paracrine mechanisms [105,106], all of which contribute to suppressing CRC tumor growth and progression.

Vitamin D deficiency is strongly linked to an increased risk of CRC, with multiple epidemiological studies emphasizing it as a significant risk factor [107]. Calcitriol helps maintain intestinal epithelial cell differentiation and integrity, essential for CRC prevention [108]. Additionally, vitamin D modulates the gut microbiome, which in CRC patients exhibits significant differences from healthy individuals. Pathogenic bacteria, such as Fusobacterium nucleatum, are more prevalent in CRC, while beneficial species, like Akkermansia muciniphila, are reduced [109,110]. Vitamin D deficiency exacerbates intestinal inflammation and promotes CRC progression, while supplementation significantly mitigates these effects [111,112].

Key anti-tumor mechanisms of vitamin D in CRC are as follows:Inhibition of cancer cell growth: calcitriol induces G1 cell cycle arrest, reducing CRC cell proliferation, and restoring sensitivity to tumor suppressors like TGF-β [113].Regulation of the Wnt/β-Catenin pathway: the Wnt/β-catenin pathway is frequently hyperactivated in CRC. Calcitriol reduces β-catenin activity and increases E-cadherin expression, stabilizing cell–cell adhesion and reducing tumor invasiveness [114].Antiangiogenesis: calcitriol inhibits angiogenesis by downregulating VEGF and NF-κB signaling, limiting the tumor’s blood supply [6].Induction of apoptosis: calcitriol promotes pro-apoptotic proteins (BAX, BAK) while inhibiting anti-apoptotic proteins (BCL-2), driving CRC cell death [2].Anti-inflammatory effects: calcitriol reduces CRC-associated inflammation by inhibiting prostaglandin synthesis, stress-activated kinases, and pro-inflammatory cytokines [67].

Vitamin D supplementation strengthens intestinal barrier function, modulates the gut microbiota, and reduces inflammation, all contributing to CRC prevention. Moreover, combining vitamin D with calcium supplementation may enhance its chemopreventive effects [115]. Calcitriol-based therapies show promise in CRC treatment, particularly by targeting the Wnt/β-catenin pathway and CYP24A1 [116].

## 7. The Role of Vitamin D in the Prevention of CRC: Immunological Mechanisms and Inflammatory Responses

The immunomodulatory properties of vitamin D are crucial for CRC prevention, as they influence both innate and adaptive immune responses. These effects play a central role in reducing inflammation and enhancing anti-tumor immunity [117,118]. Vitamin D acts on various immune cells, including T-lymphocytes, B-lymphocytes, and macrophages, all of which express vitamin D receptors. These cells also have the enzymatic ability for 1-, 24-, and 25-α hydroxylation, allowing them to synthesize the active form of vitamin D, 1,25(OH)_2_D. This enzymatic conversion is vital for modulating immune responses and inflammation [119].

One of the primary ways vitamin D influences CRC development is by reducing colonic inflammation [120]. The activation of the CYP27B1 enzyme catalyzes the conversion of vitamin D to its active form, which then triggers various anti-inflammatory pathways. These pathways mitigate the inflammatory microenvironment that accelerates CRC progression [121].

Vitamin D exerts its immunoregulatory effects through several key mechanisms. It suppresses pro-inflammatory T-helper cells, notably T-helper 1 (Th1) and T-helper 17 (Th17) lymphocytes, which play a central role in inflammatory processes and CRC development [122]. Vitamin D downregulates the production of pro-inflammatory cytokines such as interleukin-6 (IL-6), interleukin-12 (IL-12), and tumor necrosis factor-alpha (TNF-α), while promoting the secretion of anti-inflammatory cytokines, including interleukin-4 (IL-4), interleukin-5 (IL-5), interleukin-10 (IL-10), and interleukin-13 (IL-13) [123]. This shift in the Th1-Th2 balance results in a reduced inflammatory response, which is crucial for CRC prevention. Chronic inflammation is known to foster tumor cell proliferation and dissemination [124].

Vitamin D also modulates dendritic cell activity by suppressing the differentiation and activation of monocytes into dendritic cells [125]. This reduction in dendritic cell activation leads to decreased T-cell activation and antigen presentation, which could contribute to CRC prevention by preventing excessive immune activation linked to tumor progression [126]. Furthermore, vitamin D inhibits B-cell proliferation and differentiation into plasma cells, reducing autoantibody and immunoglobulin production. This action helps prevent chronic inflammation, a known contributor to carcinogenesis [127].

Additionally, vitamin D enhances macrophage function by improving chemotaxis and phagocytosis, which aids in the elimination of pathogens and malignant cells [128]. It also facilitates interactions between macrophages, dendritic cells, and T-lymphocytes, strengthening the overall immune defense against tumor formation [119,129].

Vitamin D deficiency is associated with an increased risk of CRC [107], primarily due to its negative impact on immune function and its promotion of a pro-inflammatory microenvironment, which supports tumor progression [67,130]. Chronic inflammatory diseases, such as inflammatory bowel disease, which are linked to vitamin D insufficiency, often correlate with a higher incidence of CRC [131]. Therefore, maintaining optimal vitamin D levels is vital for reducing inflammation and enhancing immune function, both of which are critical for lowering CRC risk [130].

Vitamin D supplementation has been shown to reduce CRC risk by inhibiting tumor growth and enhancing anti-tumor immunity [67,132,133]. This strategy is particularly beneficial for individuals with inflammatory bowel diseases, serving as a preventive measure against CRC. By improving immune responses, reducing inflammation, and inhibiting tumor cell proliferation, vitamin D supplementation provides enhanced protection against CRC and other malignancies [6].

## 8. The Role of Vitamin D in CRC Prevention: Mechanisms, Gut Microbiota Interaction, and Synergy with Chemotherapy and Healthy Diets

The protective effects of vitamin D are mediated through several mechanisms, one of which is ferroptosis, a form of programmed cell death associated with iron metabolism [134]. Studies indicate that 1,25(OH)_2_D suppresses colorectal cancer stem cells (CCSCs) by inducing ferroptosis [134,135]. In both in vitro and in vivo models, high levels of 1,25(OH)_2_D reduced CCSC proliferation and tumor spheroid formation by generating reactive oxygen species (ROS) and regulating SLC7A11, an antiporter involved in antioxidant cysteine uptake [135].

Vitamin D influences the sirtuin family, particularly Sirtuin 1 (SIRT1), which plays a crucial role in deoxyribonucleic acid (DNA) repair and aging processes [136]. A decrease in SIRT1 activity can inhibit cancer cell proliferation [137], while the active metabolite of vitamin D, 1,25(OH)_2_D_3_, activates SIRT1 and exerts an antiproliferative effect on colorectal cancer cells [138]. The dual role of sirtuins—as either tumor-suppressing or tumor-promoting factors—depends on their tissue-specific expression and the experimental conditions [139]. This suggests that modulating sirtuin activity could contribute to the development of personalized cancer therapies that regulate tumor-specific activities [139].

The role of SIRT1 in carcinogenesis is dose-dependent, and several studies suggest that maintaining an appropriate level of SIRT1 is critical for metabolism and tissue homeostasis [140,141]. The relationship between vitamin D and SIRT1 is particularly significant, as reduced SIRT1 activity has been linked to the pathogenesis of CRC [139]. Some studies also indicate that SIRT1 deacetylates the vitamin D receptor, thereby enhancing its activity, especially in kidney and bone cells [142,143,144]. Reduced SIRT1 activity may lead to vitamin D insensitivity. Furthermore, the VDR, when bound to its promoter, increases SIRT1 gene expression in kidney and liver cells, and vitamin D deficiency can lower both the SIRT1 level and activity [143,145,146]. However, the direct effect of vitamin D on SIRT1 activity and protein expression in colorectal cancer remains unclear [145,146].

### 8.1. Vitamin D and Its Interaction with the Gut Microbiota

The role of vitamin D extends beyond cellular mechanisms to its interaction with gut microbiota metabolites. Vitamin D affects the gut microbiome, contributing to intestinal barrier integrity, stem cell regulation, and the control of inflammatory processes [147]. Animal studies suggest that vitamin D deficiency increases CRC risk [148], while human research indicates that vitamin D supplementation reduces intestinal inflammation and promotes the enrichment of beneficial gut bacteria [149].

The study by Wyatt et al. [150], which investigated the effects of 12 weeks of supplementation with 4000 IU of D_3_ vitamin on the fecal microbiota of healthy adults, found that vitamin D supplementation significantly altered the gut microbiome composition. Specifically, Bifidobacterium, Anaerostipes, and Erysipelotrichaceae increased, while Faecalibacterium and Prevotella levels decreased, confirming the beneficial impact of vitamin D on the gut microbiota and its role in enhancing CRC resistance.

### 8.2. Vitamin D and Chemotherapeutic Synergy in Cancer

Vitamin D may also exhibit synergistic effects when combined with chemotherapeutic agents. One study demonstrated that cholecalciferol and neferine, an alkaloid derived from lotus seeds, exhibit synergistic anti-cancer effects by inhibiting CRC cell growth and metastasis. At low doses, their combination reduced cell invasion, colony formation, and migration, while downregulating the expression of N-cadherin and SNAI, key regulators of cancer cell dissemination [151,152]. SNAI, a family of transcription factors, plays an essential role in the epithelial–mesenchymal transition (EMT) process, which enhances cancer cell invasion and metastasis by suppressing E-cadherin expression.

### 8.3. The Synergistic Role of Healthy Dietary Patterns

A growing body of evidence also suggests that healthy dietary patterns, such as the Mediterranean diet, not only provide an optimal intake of vitamin D but also leverage the synergistic effects of other bioactive nutrients to mitigate the risk of multiple age-related diseases, including stroke, cardiovascular disease, and vascular cognitive impairment and dementia (VCID) [42,45,153,154,155,156,157,158,159,160,161,162,163]. The Mediterranean diet [153,161,164], characterized by a high intake of fruits, vegetables, whole grains, nuts [165], olive oil, and fish [166], provides a rich source of anti-inflammatory and antioxidant compounds, including polyphenols, omega-3 fatty acids, and essential vitamins such as vitamin D [158,163,167]. These components work in concert [168,169,170,171,172,173,174] to reduce systemic inflammation, prevent DNA damage, attenuate epigenetic aging, improve cellular and metabolic health, and ultimately protect against the progression of a spectrum of age-related diseases [175,176,177,178,179,180,181,182,183,184,185,186,187,188].

Importantly, the combined action of vitamin D and other dietary constituents [185,189] can help modulate oxidative stress, regulate immune responses, and confer anti-cancer effects, reinforcing the idea that a holistic dietary approach—rather than isolated nutrient supplementation—offers the most effective strategy for preventing age-related pathologies. Given the intricate interplay between diet, immune function, and cellular health, future research should further explore the integrative benefits of nutrient-dense, ultra-processed food-free diets [190,191], combined with vitamin supplementation, in mitigating the burden of chronic diseases associated with aging.

## 9. Association Between Serum Vitamin D Levels and CRC Outcomes

Numerous observational and prospective studies have evaluated the relationship between serum vitamin D levels and CRC outcomes. One such study [192] investigated the prognostic role of different forms of vitamin D in patients with stage I-III colorectal cancer. The study analyzed preoperative plasma 25(OH)D and vitamin D binding protein (VDBP) levels in 206 patients, demonstrating that higher free 25(OH)D levels (≥0.01–0.02 pg/mL) and biologically active 25(OH)D levels (>1.03 ng/mL) correlated with improved 5-year overall survival (OS). Free 25(OH)D was identified as an independent prognostic factor (total 25(OH)D level: high group: >29.9 ng/mL).

Facciorusso A. et al. [193] explored the role of vitamin D in the survival of CRC patients with liver metastases who underwent percutaneous radiofrequency ablation. Higher 25-hydroxy-D-vitamin levels (≥20 ng/mL) were significantly associated with longer survival and recurrence-free periods and proved to be an independent predictive factor.

Maalmi H. et al. [194] in their analysis of the DACHS cohort study data from 2910 colorectal cancer patients found that low vitamin D levels (<30 nmol/L) resulted in significantly worse survival (HR: 1.78), confirming the prognostic importance of vitamin D deficiency. Tretli S. et al. [195] found that those with the highest vitamin D levels (>46 nmol/L) had a significantly lower risk of cancer mortality (HR = 0.36).

Zgaga L. et al. [196] concluded that VDR genetic variations affect survival, and higher 25-(OH)D levels (>13.25 ng/mL) were associated with better CRC-specific and all-cause mortality. Ng K. et al. [197] found that patients with lower vitamin D levels (<20 ng/mL) had higher mortality risks, while Mezawa H. et al. [198] showed that higher 25(OH)D levels led to better survival rates, with significantly improved overall survival linked to higher levels (16–36 ng/mL).

Similar conclusions were drawn by Fedirko V. et al. [199] and Yuan et al. [200] where higher plasma 25(OH)D levels (≥24.1 ng/mL) were associated with better survival outcomes in advanced colorectal cancer patients (HR: 0.66). According to Fuchs M.A. et al. [201] higher 25-hydroxy-D-vitamin levels (≥31.5 ng/mL) were associated with better survival and lower recurrence risk in colorectal cancer patients.

An Australian cohort study [202] found that lower 25(OH)D levels (<50 nmol/L) were associated with a higher risk of colon and rectal cancer, while higher 25(OH)D levels (≥75 nmol/L) were linked to a lower risk of cardiovascular diseases and certain cancer-related mortalities. The results from the Melbourne Collaborative Cohort Study [203] indicated that higher 25(OH)D levels (53.1–121.3 nmol/L women, 68.9–201.8 nmol/L men) were associated with lower cancer-related mortality, especially for colorectal cancer, and reduced mortality from respiratory diseases (especially COPD) and gastrointestinal diseases.

Ananthakrishnan et al. [86] found that lower plasma 25(OH)D levels (<20 ng/mL) were associated with an increased risk of cancer, particularly colorectal cancer. Among IBD patients, those who were vitamin D deficient were 1.82 times more likely to develop cancer, while every 1 ng/mL increase in plasma vitamin D levels decreased the risk of colorectal cancer by 8%.

Vojdeman et al. [204] did not find any statistically significant correlation between increased vitamin D levels (≥75 nmol/L) and the incidence of breast, colorectal, urinary tract, ovarian, or corpus uteri cancers. Studies by Ordóñez-Mena J.M. et al. [205,206], Skaaby T. et al. [207], and Wong Y.Y. et al. [208] also did not show a significant association between serum 25(OH)D levels and the incidence of total or site-specific cancers. The results suggest that further research is needed to clarify the exact relationship between vitamin D and cancer.

Overall, the existing evidence suggests that increasing vitamin D levels improves CRC outcomes, particularly by enhancing post-treatment survival and reducing CRC-specific mortality. The effect of vitamin D is likely linked to the biological behavior of tumors, and future research may aim to determine the optimal maintenance of vitamin D levels for CRC patients. The following table provides a detailed overview of the association between serum 25(OH)D levels and clinical outcomes based on clinical studies conducted in colorectal cancer patients (Table 2).

## 10. The Role of Dietary Vitamin D Intake in Colorectal Cancer Prevention and Prognosis

Diet, particularly vitamin D intake, plays a crucial role in the prevention and prognosis of colorectal cancer. Over the past few decades, numerous studies have examined the relationship between dietary vitamin D intake and CRC risk [107,213]. A meta-analysis of 31 original studies reported a significant association between dietary vitamin D consumption and a reduction in CRC risk. The comparison between the highest (80 ng/mL) and lowest (10 ng/mL) dietary vitamin D intake showed a 25% reduction in risk (OR: 0.75; 95% CI: 0.67–0.85) in case–control studies [2].

The following prospective cohort studies also provide significant results supporting the hypothesis that dietary vitamin D intake reduces the risk of colon and rectal cancer. In the study by McCullough ML et al., total vitamin D intake showed an inverse association with colon cancer risk, but statistical significance was only observed for distal colon cancer (>240 IU/day vs. <90 IU/day; RR = 0.50; 95% CI: 0.24–1.04; *p*-trend = 0.04). Additionally, daily milk consumption (compared to non-consumers) was associated with a lower cancer risk in certain areas of the colon, especially proximal colon cancer (RR = 0.68; 95% CI: 0.42–1.09; *p*-trend = 0.06) [214].

The Nurses’ Health Study [215] reported that women with the highest vitamin D intake group had a 58% reduced risk of colon cancer (HR: 0.42; 95% CI: 0.19–0.91) compared to those with lower intake (<76 IU/day vs. >477 IU/day). Consistently higher vitamin D intake (>550 IU/day vs. <76 IU/day) showed an even stronger protective effect (HR: 0.33; 95% CI: 0.16–0.70) [215]. In the Iowa Women’s Health Study, after a 5-year follow-up, those in the highest calcium and vitamin D intake categories (<159 IU/day vs. >618 IU/day) had about half the risk of colon cancer compared to those in the lowest intake category. Dairy consumption showed a similar trend, although it was statistically less significant (*p* < 0.05) [216]. The most significant reduction was observed in rectal cancer: the RR for the highest vitamin D intake group was 0.42 (95% CI: 0.19–0.91; vitamin D > 618 IU/day). According to the study, vitamin D may have a stronger association with the reduction in colorectal cancer risk than calcium, particularly in the case of rectal cancer.

In Kearney J et al.’s study, higher vitamin D intake (810 IU/day) initially showed an inverse association with colon cancer risk (RR = 0.54; 95% CI: 0.34–0.85; *p* = 0.0006), but after adjusting for multiple variables, this relationship weakened (RR = 0.66; 95% CI: 0.42–1.05; *p* = 0.02) [217]. Zheng W et al. demonstrated that the combined effect of calcium and vitamin D strongly reduced the risk of rectal cancer, particularly in those with higher calcium and vitamin D intake (Ca > 1278.7 mg/day + vitamin D > 337 IU/day) [218].

In the Spanish PREDIMED study, the fully adjusted model (model 4) showed that those in the highest vitamin D intake quartile had a 45% reduced risk of CRC compared to the lowest quartile (HR: 0.55; 95% CI: 0.30–1.00; *p*-trend = 0.072), although this association was not statistically significant. However, for colon cancer, a significant 56% inverse relationship was found (HR: 0.44; 95% CI: 0.22–0.90). After excluding participants who were taking vitamin D and/or calcium supplements, the CRC risk reduction increased to 48% (HR: 0.52; 95% CI: 0.28–0.96), with a 59% reduction in colon cancer risk (HR: 0.41; 95% CI: 0.12–0.85) [219].

In the Danish “Diet, Cancer and Health” Study, total vitamin D intake did not show a significant relationship with colon cancer risk (IRR = 1.01; 95% CI: 0.87–1.18; vitamin D 2.3 μg/day vs. 10.2 μg/day), but certain genetic variants (CYP2R1 and GC/rs4588) may enhance the protective effect of vitamin D. The combined genetic risk score (GRS) with two risk alleles showed a 10% reduction in colon cancer risk with 3 µg/day of vitamin D intake (IRR = 0.90; 95% CI: 0.81–0.99) [220].

In Kesse E et al.’s research, increasing vitamin D intake did not show a significant effect on colon cancer risk, although the risk of adenomas was reduced in the high vitamin D intake group (<1.72 µg/day vs. >3.23 µg/day) [221].

Nakano S et al. found an inverse relationship between vitamin D intake and CRC risk in the presence of high VDR expression in stroma (highest third vs. lowest third: HR 0.46 [0.23–0.94]; *p*-trend = 0.03), suggesting that high vitamin D intake (534.6 IU/day vs. 154.1 IU/day) may reduce colon cancer risk, but this is mainly dependent on stromal VDR expression [222].

In Garland C et al.’s study, after nearly two decades of follow-up, it was found that dietary vitamin D and calcium intake were significantly inversely associated with colon cancer risk. In the group with the highest vitamin D–calcium index, the colon cancer risk was 14.3/1000, compared to 38.9/1000 in the lowest group (vitamin D: 75–208 vs. 2–30 IU/1000 kcal/day). The association remained significant after adjusting for age, smoking, BMI, alcohol consumption, and fat intake [223].

Studies by Ishihara J et al. [224], Järvinen R et al. [225], and Terry P et al. [226] found no significant association between vitamin D intake and colorectal or rectal cancer risk. In the following table, we present the detailed results of the studies (Table 3).

## 11. Ensuring Adequate Vitamin D Intake and Dietary Recommendations

To ensure adequate vitamin D intake, it is recommended to consume vitamin D-rich foods, such as fatty fish (salmon, mackerel, tuna), egg yolks, and vitamin D-fortified dairy products [227]. However, the Western diet, which often contains a high proportion of ultra-processed foods, added sugars, trans fats, and red or processed meats, contributes to the development of chronic inflammatory conditions and worsens vitamin D deficiency [228]. These foods are not only nutrient-poor but also hinder the absorption and utilization of vitamin D. Moreover, reduced sun exposure, sedentary lifestyles, and habits related to UV protection further diminish the body’s ability to maintain adequate vitamin D levels [229].

To reduce the risk of colorectal cancer, it is crucial to follow a healthy, nutrient-dense diet that provides sufficient vitamin D, as well as foods rich in calcium and fiber [115]. In addition to proper nutrition, regular physical activity, weight management, and moderate sun exposure also play a role in colorectal cancer prevention [230]. If vitamin D intake from natural sources is insufficient, supplementation may be necessary, particularly for at-risk groups such as the elderly, individuals with darker skin tones, and those with limited sun exposure [231]. The following table lists foods rich in vitamin D and their respective vitamin D content (International Units, IU), providing assistance in ensuring an adequate daily intake of vitamin D (Table 4).

## 12. Vitamin D Supplementation and Its Relationship with CRC

In the 2022 systematic meta-analysis by Lopez-Caleya et al. [233], the effects of calcium and vitamin D intake on CRC risk were specifically investigated. According to the results, every 100 IU/day increase in vitamin D intake reduced the CRC risk by 4% (OR: 0.96; 95% CI: 0.93–0.98). This association was observed regardless of gender, tumor location, or geographical region [233]. Another systematic meta-analysis found that vitamin D supplementation (500–2000 IU/day) reduced CRC incidence (OR = 0.87; 95% CI: 0.82–0.92) and improved long-term survival in patients (HR = 0.91; 95% CI: 0.83–0.98) [234].

The SUNSHINE Trial [235] aimed to investigate whether adding high-dose vitamin D3 to standard chemotherapy improves the outcome of metastatic colorectal cancer. The high-dose group received an initial dose of 8000 IU of vitamin D, followed by 4000 IU/day, while the standard-dose group received 400 IU/day throughout the treatment. The results indicated that patients receiving high-dose vitamin D3 had an average of two months longer progression-free survival (13 months vs. 11 months), which was a statistically significant difference. The treatment was deemed safe, as the frequency and severity of side effects were similar in both groups. These results suggest that vitamin D3 may have a potentially beneficial effect, but further larger studies are needed to confirm this.

In the Iowa Women’s Health Study, participants who took calcium and vitamin D supplements had a statistically significant 15% lower risk of CRC compared to those who did not take any supplements (HR = 0.85; 95% CI: 0.75–0.97; total vitamin D, IU/day: 199 vs. 656; total calcium, mg/day: 443 vs. 1.957) [236].

In the study by Park SY et al. [237], a significant inverse relationship was found in men between vitamin D intake and CRC risk (HR = 0.72; 95% CI: 0.51–1.00), suggesting that higher vitamin D intake reduces CRC risk. No significant association was found in women (men 335 IU/day; women 340 IU/day).

The VITAL trial also examined the relationship between vitamin D and CRC, with subgroup analysis revealing differences based on body mass index (BMI). Among normal-weight participants who received vitamin D (2000 IU/day), a lower CRC incidence was observed compared to those who received a placebo. For participants with a BMI below 27.1, the HR for any type of invasive cancer was 0.86 (95% CI: 0.75–0.99), whereas for those with a BMI ≥ 27.1, it was 1.08 (95% CI: 0.94–1.24), suggesting that BMI, or nutritional status, may influence the effect of vitamin D [238].

In the Women’s Health Initiative study [239], 36,282 postmenopausal women participated, of whom 18,176 women received 500 mg of calcium and 200 IU of vitamin D3 daily, while the other group received a placebo. After 7 years of follow-up, no significant differences were found in the incidence of colorectal cancer between the group receiving calcium and vitamin D3 and the placebo group.

In another study, the effect of postoperative vitamin D supplementation on improving survival in gastrointestinal cancer patients was investigated [240]. Of the 439 participants, 251 patients received 2000 IU of vitamin D3 daily, while 166 patients received a placebo. The results indicated that vitamin D did not result in a significant improvement in relapse-free survival, although better outcomes were observed in patients whose initial 25(OH)D levels were between 20 and 40 ng/mL.

Another study [241] evaluated the effect of daily 2000 IU of cholecalciferol on survival in metastatic colorectal cancer (mCRC). Seventy-two mCRC patients with 25(OH)D levels < 75 nmol/L were randomized to receive either standard chemotherapy or chemotherapy with 2000 IU of vitamin D daily. The results showed no statistically significant differences in overall survival or progression-free survival between the vitamin D supplementation group and the control group.

In a large U.S. cohort study [242], 39,876 women aged 45 and older were followed for 10 years for their calcium and vitamin D intake. The median daily intake was 882 mg of calcium and 271 IU of vitamin D, with dietary intake of 705 mg of calcium and 205 IU of vitamin D. The data did not support an association between calcium and vitamin D intake and the risk of colorectal and rectal cancer.

Although not all relevant studies yielded statistically significant results, the available evidence suggests that vitamin D, whether from dietary sources or supplements, may reduce the risk of colorectal carcinoma incidence (Table 5).

## 13. Recommended Intakes of Vitamin D

Raising 25-hydroxyvitamin D concentrations above 30 ng/mL (75 nmol/L) can significantly reduce the risk of common causes of death, such as heart disease, cancer (particularly colorectal, breast, and prostate cancers), stroke, high blood pressure, chronic respiratory diseases, Alzheimer’s disease, diabetes, kidney diseases, and COVID-19, as well as help prevent mortality associated with these conditions [166,245].

Vitamin D supplementation is the most effective method for increasing 25(OH)D concentrations, as it can be carried out year-round and under controlled conditions. Research indicates that a daily dose of 2000 IU (50 µg) is the optimal minimum dose for adults of normal weight, allowing them to reach a 30–40 ng/mL level with minimal safety concerns [246]. This dose can be taken daily, weekly (15,000 IU), or monthly (60,000 IU), with compliance typically being better with weekly or monthly dosing. Low-dose supplementation is also recommended for individuals seeking to reach the optimal level over a longer period, though it may take several months to achieve a steady-state concentration [247]. For this reason, it is recommended to take larger “bolus” doses during the first one to two weeks to reach the optimal vitamin D level more quickly [248].

The recommended daily doses of vitamin D in Hungary were revised and increased based on the 2022 consensus statement, “Hungarian Consensus Recommendation on the Role of Vitamin D in the Prevention and Treatment of Diseases”. The updated reference values vary by age group (Table 6) [4].

## 14. Vitamin D Supplementation and the Development of Adenomas and Polyps

The findings from the Nurses’ Health Study, the Nurses’ Health Study 2, and the Health Professionals Follow-up Study [249] confirm that vitamin D plays a crucial role in reducing the risk of colorectal cancer. Lower levels of vitamin D (<50 nmol/L) have been associated with both synchronous polyps (SPs) and conventional adenomas. The study also highlights that vitamin D intake (415 ± 214 IU/day) provides stronger protection against conventional adenomas, particularly in the distal colon, compared to the proximal colon. While the research also emphasized the role of other nutritional factors such as calcium and folate intake, the effect of vitamin D appears particularly significant in the prevention of SPs and conventional adenomas.

In a cohort study by Sutherland RL et al. [250], based on data from 1409 Canadian participants, vitamin D supplementation (600 IU/day) was found to reduce the likelihood of colorectal polyp and HRAP (high-risk adenomatous polyps) formation. The average age of the participants was 60 years (SD = 6). According to the results, vitamin D supplementation reduced the probability of polyp development by 33% (adjusted odds ratio [ORadj] = 0.67; 95% CI: 0.51–0.88) and the risk of HRAPs by 43% (ORadj = 0.57; 95% CI: 0.33–0.96). These findings align with earlier studies suggesting that vitamin D supplementation may play a role in the prevention of colorectal polyps.

A study by Ahearn TU et al. [251] aimed to investigate the effects of calcium and vitamin D on the expression of adenomatous polyposis coli (APC), β-catenin, and E-cadherin in the normal-appearing mucosa of sporadic colorectal adenoma patients. The clinical trial involved 92 patients who received 2000 mg/day calcium and/or 800 IU/day vitamin D for 6 months. The results showed that vitamin D and calcium supplementation increased APC expression, decreased β-catenin levels, and increased E-cadherin expression, which is consistent with a reduction in colorectal cancer risk. This research supports the potential chemopreventive role of calcium and vitamin D in colorectal cancer prevention.

Kwan AK et al. [252] explored the effects of 1000 IU/day vitamin D and 1200 mg/day calcium supplements on the expression of DNA repair proteins, such as MSH2 and TGFα/TGFβ1 biomarkers, in the normal-appearing colonic mucosa. Although the supplements did not show statistically significant effects, vitamin D increased the expression of MSH2 and TGFβ1, while the TGFα/TGFβ1 ratio decreased, indicating a reduction in the risk of colorectal carcinogenesis.

Crockett SD et al. [253], Baron JA et al. [254], and Song M et al. [255] found no significant effect of vitamin D or calcium supplementation on the incidence of colorectal adenomas and serrated polyps in three separate randomized controlled trials (Table 7).

In summary, vitamin D plays a crucial role in the prevention and treatment of colorectal cancer by reducing inflammation, regulating immune responses, promoting cell death, and inhibiting tumor angiogenesis. Numerous observational and prospective studies have examined the relationship between serum vitamin D levels and colorectal cancer outcomes, and it has been found that higher serum 25(OH)D levels are associated with better survival and lower cancer-specific mortality. Low vitamin D levels are linked to worse survival outcomes, while higher levels serve as an independent prognostic factor. Some studies also suggest that vitamin D supplementation improves survival, particularly in patients with advanced-stage colorectal cancer. Regarding vitamin D supplementation and the risk of adenomas and polyps, research indicates that vitamin D may help reduce the risk of colorectal polyps, especially high-risk adenomas.

Doctors should be aware of the role of vitamin D in the prevention and treatment of major diseases and inform patients about its benefits. The 25-hydroxyvitamin D levels should be measured in patients at higher risk of deficiency, or it may be advisable to administer 1000–4000 IU of vitamin D3 daily. It is recommended that all pregnant women take 2000–4000 IU of vitamin D3 during pregnancy and breastfeeding. Supplementation guidelines for vitamin D may vary on an individual basis, and the optimal dosage depends on the individual’s current health status, age, and vitamin D levels.

## 15. Critical Remarks and Limitations

Although numerous studies support the protective effects of vitamin D, the exact mechanisms through which it exerts its effects on CRC prevention and treatment remain unclear. Most research focuses on epidemiological and mechanistic approaches, while large-scale, randomized clinical trials that could provide clearer answers are still lacking.

Further studies are needed to determine the optimal dosage and formulation of vitamin D supplementation and to clarify the role of genetic and lifestyle factors in modulating its anti-cancer effects. Additionally, more research should explore the direct interaction of vitamin D with CRC pathophysiology and investigate its synergistic potential with other therapeutic agents.

## 16. Conclusions

In summary, vitamin D plays a crucial role in both CRC prevention and treatment. It influences cancer cell proliferation, inflammation, cell death mechanisms, and the activity of the Sirtuin protein family. Recent research highlights that maintaining optimal vitamin D levels can help reduce CRC risk and improve treatment outcomes. However, further studies are needed to elucidate the precise mechanisms of action and optimize the therapeutic application of vitamin D in CRC management.

## Figures and Tables

**Figure 1 nutrients-17-01351-f001:**
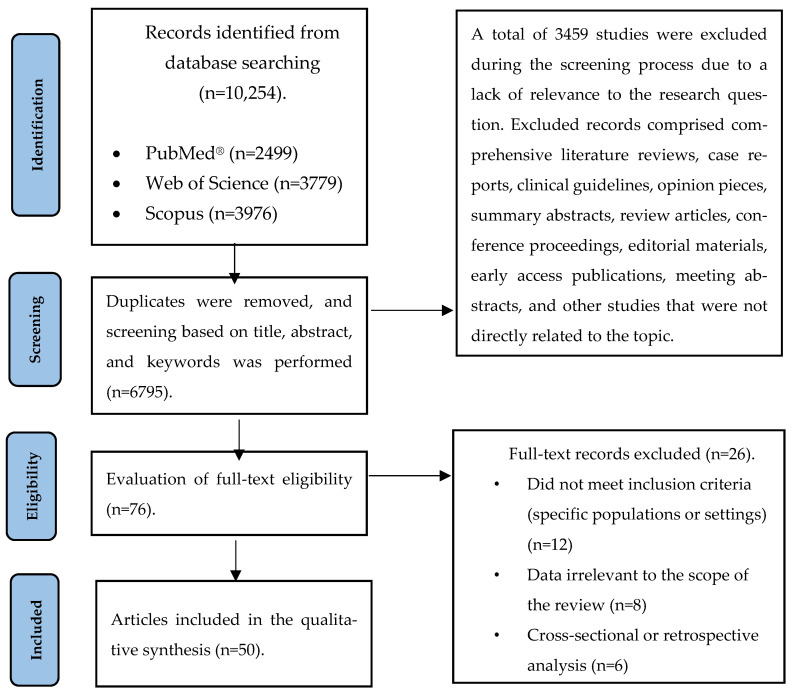
Flowchart illustrating the selection process of the included articles.

**Table 1 nutrients-17-01351-t001:** PICO framework for literature review on vitamin D and colorectal cancer.

PICO	Criteria
Population	Adult patients who are either healthy, have documented vitamin D deficiency, or have been diagnosed with CRC, as well as early-stage lesions such as adenomas or polyps.
Intervention	Vitamin D intake or supplementation and its effects on CRC development, immune response, and inflammatory processes.
Comparison	Individuals with vitamin D deficiency or those not receiving vitamin D supplementation.
Outcome	Incidence of colorectal cancer/adenomas/polyps, levels of immunological markers, concentrations of inflammatory factors, tumor progression, and overall disease course.

**Table 2 nutrients-17-01351-t002:** Summary of observational prospective clinical studies on serum vitamin D levels and CRC outcomes.

Study	Design	Mean Follow-Up	Country	Sample Size	Average Age (Year)	Sex Male/ Female (%)	CRC Stage	Main Results (HR, 95% CI)
Yang L et al. [192]	Prospective cohort	45 months	China	206	63	63.5/36.5	Stage I–III CRC	Higher free 25(OH)D levels (≥0.01–0.02 pg/mL) were identified as an independent factor for improved overall survival (HR = 0.442, 95% CI = 0.238–0.819, *p* < 0.010)
Facciorusso A et al. [193]	Prospective cohort	72 months	Italy	143	68	71.3/28.7	CRC with liver metastases	HR based on 25(OH)D levels (≥20 ng/mL) HR: 0.35 (95% CI: 0.21–0.59), *p* < 0.001
Maalmi H et al. [194]	Prospective cohort	4.8 years	Germany	2910	69	60/40	Stage I–IV CRC	All-cause mortality: HR = 1.78 (95% CI: 1.39–2.27); CRC-specific mortality: HR = 1.65 (95% CI: 1.24–2.21); 25(OH)D < 30 nmol/L
Tretli S et al. [195]	Prospective cohort	30 years	Norway	658 (CRC: 52)	59.1	61.5/38.5	Stage I–IV CRC (with and without metastases)	HR = 0.36 (95% CI: 0.27–0.51); 25(OH)D < 46 nmol/L
Zgaga L et al. [196]	Prospective cohort	12,323 person-years of follow-up	Ireland	1598	62.5	58/42	Stage I-III CRC	CRC-specific mortality: HR = 0.68 (95% CI: 0.50–0.90); all-cause mortality: HR = 0.70 (95% CI: 0.55–0.89); 25(OH)D ≥ 13.25 ng/mL
Ng K et al. [197]	Prospective cohort	5.1	Ireland	515	61	59/41	Unresectable metastatic colorectal cancer	No significant association between plasma 25(OH)D levels and overall survival (HR = 0.94, *p* trend = 0.55); 25(OH)D ≥ 33 ng/mL
Mezawa H et al. [198]	Prospective cohort	32.4 months	Japan	257	65 ± 12	65/35	Stage I–IV CRC	Higher 25(OH)D levels (≥30 ng/mL) are associated with better overall survival (HR, 0.91; 95% CI, 0.84–0.99, *p* = 0.027)
Fedirko V et al. [199]	Prospective cohort	73 months	Europe (EPIC Study)	1202	62.1 (7.6)	49.5/50.5	Stage I–IV CRC	Higher prediagnostic 25(OH)D levels (≥76.9 nmol/L) are associated with lower CRC-specific mortality (HR 0.69, 95% CI: 0.50–0.93) and overall mortality (HR 0.67, 95% CI: 0.50–0.88)
Yuan C et al. [200]	Prospective cohort	5.6 years	USA	1041	59 (12)	58/42	Advanced or metastatic CRC	OS: HR = 0.66 (95% CI: 0.53–0.83); PFS: HR = 0.81 (95% CI: 0.66–1.00); 25(OH)D ≥ 24.1 ng/mL
Fuchs MA et al. [201]	Prospective observational study	3.5 (0.2–9.9) months	USA	1016	60.4	56/44	Stage III CRC	DFS: HR = 0.62 (95% CI: 0.44–0.86), Ptrend = 0.05. OS: HR = 0.55 (95% CI: 0.38–0.80), Ptrend = 0.0004; 25(OH)D: 30.1–36.4 ng/mL
Zhu K et al. [202]	Prospective cohort	20 years	Australia	3818	25–84	43/57	Colorectal cancer	CRC risk: Low 25(OH)D < 50 nmol/L associated with higher CRC risk (HR 1.62, 95% CI 1.04–2.53)
Heath AK et al. [203]	Case–cohort study	14 years	Australia	2923	61.3	55.2/44.8	Colorectal cancer	Colorectal cancer: HR = 0.75 (95% CI 0.57–0.99), women (25(OH)D: 53.1–121.3 nmol/L) HR = 0.63 (95% CI 0.40–1.01), men (25(OH)D: 68.9–201.8 nmol/L) HR = 0.82 (95% CI 0.58–1.14)
Vojdeman FJ et al. [204]	Observational cohort study	10 years	Denmark	1108	48.8	65.3/4.7	Colorectal cancer (rectosigmoid cancer)	HR: 0.98 (95% CI: 0.96–1.00), *p* = 0.1; 25(OH)D < 30 nmol/L
Ordóñez-Mena JM et al. [205]	Cohort ESTHER/TROMSØ/EPIC-Elderly	12 years	Germany/Norway/Greece, Denmark, Netherlands, Spain, Sweden	616	63	42.9/57.1	Colorectal cancer	ESTHER: HR 0.99 (0.60–1.65); TROMSØ: HR 1.33 (0.73–2.44); EPIC-Elderly: OR 1.24 (0.64–2.42); meta-analysis: RR 1.15 (0.82–1.61), *p* = 0.74; 25(OH)D < 50 nmol/L
Ordóñez-Mena JM et al. [206]	Prospective cohort	8 years	Germany	9949	50–74	42/58	Stage I–IV CRC	HR for Q1 (lowest 25(OH)D quartile): 1.33 (1.06–1.68) in men, 0.95 (0.75–1.20) in women. Protective effect for obese individuals: HR: 0.65 (0.48–0.90) in the lowest quartile of 25(OH)D < 30 nmol/L
Skaaby T et al. [207]	Prospective cohort	11.3 years	Denmark	12,204	≥55	49.9/50.1	Colorectal cancer	HR = 0.95 (95% CI, 0.88–1.02); 25(OH)D < 50 nmol/L
Wong YY et al. [208]	Prospective cohort	6.7 ± 1.8 years	Australia	4208	70–88	100% Male	Colorectal cancer	HR = 0.88 (95% CI, 0.55–1.40); 25(OH)D < 50 nmol/L
Cooney RV et al. [209]	Prospective cohort	8.03	USA	368	<85	58.7/41.3	Stage I–IV CRC	HR = 0.98 (95% CI: 0.57–1.67); *p*-value for trend = 0.92, indicating no significant association between 25(OH)D levels and CRC-specific mortality; 25(OH)D > 30.8 ng/mL
Ng K et al. [210]	Prospective cohort study	Up to 14 years (1991–2005)	USA	304	68.4	47/53	Stage I–IV CRC	Higher prediagnosis 25(OH)D ≥ 40.0 ng/mL levels are associated with lower overall mortality (HR = 0.52, 95% CI: 0.29–0.94, *p* trend = 0.02) and a trend toward lower CRC-specific mortality (HR = 0.61, 95% CI: 0.31–1.19)
Ananthakrishnan AN et al. [211]	Observational cohort study	11 years	USA	2809	46 (IQR 32–60)	39/61	Stage I–IV CRC (with and without metastases)	Each 1 ng/mL increase in 25(OH)D reduced CRC risk by 8% (OR = 0.92, 95% CI: 0.88–0.96); median 25(OH)D: 26 ng/mL
Cheney CP et al. [212]	Population-based prospective cohort	7 years	Germany	2003	59.7 (SD 11.8)	62.3/37.7	Colorectal cancer	HR 0.97 (95% CI: 0.88–1.07) for CRC risk per 1 ng/mL increase in 25(OH)D (<20 ng/mL)

Abbreviations: %: percent; 25(OH)D: 25-hydroxyvitamin D; CI: Confidence Interval; CRC: Colorectal Cancer; DFS: Disease-Free Survival; EPIC: European Prospective Investigation into Cancer and Nutrition; ESTHER: Epidemiological Study on the Chances of Prevention; Early Detection; and Optimized Therapy of Chronic Diseases in the Elderly; HR: Hazard Ratio; IQR: Interquartile Range; OR: Odds Ratio; OS: Overall Survival; PFS: Progression-Free Survival; RR: Relative Risk; SD: Standard Deviation; TROMSØ: Tromsø Study

**Table 3 nutrients-17-01351-t003:** Summary of clinical studies on the association between dietary vitamin D intake and colorectal cancer risk.

Study	Design	Mean Follow-Up	Country	Sample Size	Average Age (Year)	Sex Male/ Female (%)	CRC Stage	Main Results (HR, 95% CI)
McCullough ML et al. [214]	Prospective cohort	5 years	USA	127,749	62.8	48/52	Incident CRC cases (421 men, 262 women)	Vitamin D intake (>240 IU/day vs. <90 IU/day): RR = 0.71 (95% CI: 0.51–0.98) in men, *p* trend = 0.02.
Martínez ME et al. [215]	Prospective cohort	12 years	USA	89,448	30–55	100% Female	Colorectal adenocarcinoma (colon and rectal cancer)	After excluding milk intake changers: total vitamin D (<76 IU/day vs. >477 IU/day): 0.42 (0.19–0.91); consistent high total vitamin D: 0.33 (0.16–0.70).
Bostick RM et al. [216]	Prospective cohort	5 years	USA	35,216	55–69	100% Female	Colorectal cancer	Vitamin D (<159 IU/day vs. >618 IU/day): 0.54 (0.35–0.84) (age-adjusted), 0.73 (0.45–1.18) (multivariate-adjusted).
Kearney J et al. [217]	Prospective cohort	6 years	USA	47,935	40–75	100% Male	Colon cancer	Total vitamin D (810 IU/day): RR = 0.66 (95% CI: 0.42–1.05). Dietary vitamin D: RR = 0.88 (95% CI: 0.54–1.42).
Zheng W et al. [218]	Cohort Study	9 years	USA	34,702	55–69	100% Female	Colorectal cancer	Vitamin D intake: RR 1.00, 0.71, 0.76 (*p* = 0.20); highest intake of both calcium and vitamin D (Ca > 1278.7 mg/day + vitamin D > 337 IU/day): RR 0.55, 95% CI 0.32–0.93 (45% reduced risk).
Hernández-Alonso P et al. [219]	Cohort study, observational	6 years	Spain	7216	67	57/43	Incident CRC and colon cancer	Colon cancer: 0.44 (0.22–0.90), *p* for trend = 0.032 (significant). The highest vitamin D intake was 618 IU/day.
Kopp TI et al. [220]	Nested case–cohort	15 years	Denmark	920 cases/1743 controls	58	56/44	Colorectal cancer	IRR: 1.01 (0.87–1.18) (vitamin D: 2.3 μg/day vs. 10.2 μg/day). Not significant.
Kesse E et al. [221]	Prospective cohort	3.7 years	France	67,484	52.7 (6.6)	100% Female	Colorectal cancer	No significant association with vitamin D (<1.72 µg/day vs. >3.23 µg/day).
Nakano S et al. [222]	Prospective study	15 years	Japan	22,743	61.3 (6.2)	46.7/53.3	Various (82.3% high VDR in tumors, 12.1% high VDR in stroma)	HR 0.46 (0.23–0.94) (534.6 IU/day vs. 154.1 IU/day).
Garland C et al. [223]	Prospective cohort study	19 years	USA	1954	50 (4)	100% Male	Colorectal cancer	The risk of CRC in the highest quartile of Vitamin D and calcium intake (75–208 vs. 2–30 IU/1000 kcal/day) was 14.3/1000, compared to 38.9/1000 in the lowest quartile.
Ishihara J et al. [224]	Prospective Cohort Study	9.5 years	Japan	74,639	50.8± 7.5	47/53	All stages	No significant association (the highest D-vitamin intake was 21.0 ± 7.4 μg/day).
Järvinen R et al. [225]	Prospective cohort study	24 years	Sweden	9959	53,7	60/40	Colon and rectal cancer	No significant association with vitamin D: 3.8 µg/day.
Terry P et al. [226]	Cohort study, observational	11.3 years	Sweden	61,463	53	100% Female	Colon and rectal CRC	Rate ratio (4th vs. 1st quartile): 1.05 (95% CI = 0.83–1.33; vitamin D intake: 2.9 µg/day (lowest quartile) to 3.7 µg/day (highest quartile).

Abbreviations: CRC: colorectal cancer; HR: hazard ratio; RR: relative risk; CI: confidence interval; IRR: incidence rate ratio; VDR: vitamin D receptor.

**Table 4 nutrients-17-01351-t004:** Foods rich in vitamin D.

Food	Vitamin D Content (IU)
Cow’s milk	3–40/L
Fortified milk/infant formulas	400/L
Fortified orange juice/soy milk/rice milk	400/L
Butter	35/100 g
Margarine, fortified	60/tablespoon
Yogurt (normal, low fat, or nonfat)	89/100 g
Cheddar cheese	12/100 g
Parmesan cheese	28/100 g
Swiss cheese	44/100 g
Cereal fortified	40/serving
Tofu fortified (1⁄5 block)	120
Fresh shiitake mushrooms	100/100 g
Dried shiitake mushrooms (non-radiated)	1660/100 g
Egg yolk	20–25 per yolk
Shrimp	152/100 g
Calf liver	15–50/100 g
Canned tuna/sardines/salmon/mackerel in oil	224–332/100 g
Canned pink salmon with bones in oil	624/100 g
Cooked salmon/mackerel	345–360/100 g
Atlantic mackerel (raw)	360/100 g
Atlantic herring (raw)	1628/100 g
Smoked herring	120/100 g
Pickled herring	680/100 g
Codfish (raw)	44/100 g
Cod liver oil	175/g; 1360/tablespoon

Source [232]. 1 IU of vitamin D (either cholecalciferol [D_3_] or ergocalciferol [D_2_]) = 0.025 micrograms (µg). 1000 IU = 25 µg of vitamin D. International Unit: A unit of measurement for vitamins and other biologically active substances.

**Table 5 nutrients-17-01351-t005:** Summary of studies on vitamin D and colorectal cancer risk.

Study	Design	Mean Follow-Up	Country	Sample Size	Average Age (Year)	Sex Male/ Female (%)	CRC Stage	Main Results (HR, 95% CI)
Ng K et al. [235]	RCT	22.9 months	USA	139	56	57/43	Metastatic	HR for PFS: 0.64 (95% CI, 0–0.90, *p* = 0.02). High-dose: 8000 IU initial, 4000 IU/day; standard-dose: 400 IU/day.
Um CY et al. [236]	Prospective cohort	26 years	USA	35,221	55–69	100% Female	Overall and distal CRC	HR = 0.85; 95% CI, 0.75–0.97. Total vitamin D: 656 IU/day; total calcium: 1957 mg/day.
Park SY et al. [237]	Cohort Study	8 years	Los Angeles, California	191,011	58.1	45/55	Invasive CRC	Total vitamin D intake: Men: RR = 0.72, 95% CI: 0.51–1.00; *p* = 0.03. No significant association in women. (Men: 335 IU/day; women: 340 IU/day).
Manson JE et al. [238]	RCT	5.3 years	USA	25,871	67.1	49/51	All cancer types	No significant reduction in invasive cancer HR: 0.96 (95% CI: 0.88–1.06; 2000 IU/day).
Wactawski-Wende J et al. [239]	RCT	7 years	USA	36,282	50–79	100% Female	Invasive CRC	HR = 1.08 (95% CI: 0.86–1.34), *p* = 0.51; (500 mg of calcium and 200 IU of D3 vitamin).
Urashima M et al. [240]	RCT	3.5 years	Japan	417	66	66/34	Stage I-III CRC	Relapse-free survival: HR = 0.76 (95% CI, 0.50–1.14; *p* = 0.18). Overall survival: HR = 0.95 (95% CI, 0.57–1.57; *p* = 0.83); Vit D: 2000 IU/day.
Antunac Golubić Z et al. [241]	RCT	46 months	Croatia	71	63 (56–71)	61.8/40.5	Metastatic CRC	No significant difference in OS or PFS (HR = 1.0064, 95% CI = 0.3882–2.609, *p* = 0.9895); Vit D: 2000 IU/day.
Lin J et al. [242]	Prospective Cohort Study	10 years	USA	36,976	54	100% Female	Colorectal cancer	Total vitamin D: 1.34 (95% CI: 0.84, 2.13; *p* for trend = 0.08). No significant association with CRC risk. The median daily intake was 882 mg of calcium and 271 IU of vitamin D.
Serrano D et al. [243]	Randomized Phase II Trial	2.6 years	Italy	74	62	47/53	Stage II-III CRC	WCRF adherence significantly decreased the risk of events (HR = 0.41, 95% CI: 0.18–0.92, *p* = 0.03). No significant difference with vitamin D supplementation alone (2000 IU/day).
Paulsen EM et al. [244]	Prospective cohort study	6 years	Norway	95,416	56	Female 100%	Proximal colon cancer, distal colon cancer, rectal cancer	CRC: 5 μg increase in vitamin D intake: HR = 0.97 (95% CI: 0.93, 1.01). Proximal colon cancer: 10–19 μg intake: HR = 0.73 (95% CI: 0.57, 0.94; high intake vit D ≥ 20 µg/day).

Abbreviations: CRC: colorectal cancer; HR: hazard ratio; RCT: randomized controlled trial; PFS: progression-free survival; OS: overall survival; RR: relative risk; WCRF: World Cancer Research Fund.

**Table 6 nutrients-17-01351-t006:** Recommended daily vitamin D doses for the prevention of vitamin D deficiency in Hungary.

Age Group	Recommended Daily Dose	Upper Safe Daily Intake Limit
Under 1.5 years	400–500 IU	1000 IU
Children (1.5–6 years)	400–500 IU	1000 IU
High-risk children (1.5–6 years)	1000 IU	2000 IU
Children (above 6 years)	1000 IU	2000 IU
Adults	2000 IU	4000 IU

Source [4] 1 IU = 0.025 μg vitamin D.

**Table 7 nutrients-17-01351-t007:** Summary of clinical trials on vitamin D and colorectal adenomas/polyps.

Study	Design	Mean Follow-Up	Country	Sample Size	Average Age (Year)	Sex Male/ Female (%)	Stage	Main Results (HR, 95% CI)
He X et al. [249]	Prospective cohort	20 years	USA	141,143	60.2	20/80	SPs, conventional adenomas	Higher intake of vitamin D (415 ± 214 IU/day) and marine omega-3 fatty acid (0.25 ± 0.20 g) were associated with lower risk.
Sutherland RL et al. [250]	Observational study	14.7 years	Canada	1409	60 ± 6	54.7/45.3	Early-stage CRC (colorectal polyps)	ORadj = 0.67 (95% CI: 0.51–0.88); reduced odds of HRAPs: ORadj = 0.57 (95% CI: 0.33–0.96; vit D: 600 IU/day).
Ahearn TU et al. [251]	RCT	6 months	USA	92	61	70/30	Colorectal adenoma	Calcium intake: 2000 mg/day Vitamin D intake: 800 IU/day Combination: β-catenin −11% (*p* = 0.20), E-cadherin +51% (*p* = 0.08).
Kwan AK et al. [252]	RCT	1 year	USA	104	59	46/54	Colorectal adenoma (early-stage)	1000 IU/day vitamin D and 1200 mg/day calcium supplements vs. calcium RR: 0.56 (0.26, 1.20).
Crockett SD et al. [253]	RCT	5 years	USA	2813	58.1	63/37	Polyps, SSA/Ps (sessile serrated adenoma), and traditional serrated adenoma	During the treatment phase, neither calcium (1200 mg/day) nor vitamin D (1000 IU/day) had an effect on the incidence of SSA/Ps.
Baron JA et al. [254]	RCT	5 years	USA	2259	58.2 ± 7.0	85.5/14.5	SPs, Conventional adenomas	HR (95% CI): 0.99 (0.89–1.09) for vitamin D, 0.93 (0.80–1.08) for D + calcium (1000 IU/day + 1200 mg/day).
Song M et al. [255]	RCT	5.3 years	USA	25,871	67.1 ± 7.1	49.4/50.6	Adenoma, serrated polyps	D-vitamin supplementation (2000 IU/day) not associated with colorectal adenomas or serrated polyps risk.

Abbreviations: SPs: sessile polyps; CRC: colorectal cancer; SSA/Ps: sessile serrated adenomas/polyps; HRAPs: high-risk adenomatous polyps; RCT: randomized controlled trial; ORadj: adjusted odds ratio; 95% CI: 95% confidence interval.

## Data Availability

Data sharing is not applicable to this article as no new data were created or analyzed in this study.

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
