# Peer review of "Vitamin D and Colorectal Cancer Prevention: Immunological Mechanisms, Inflammatory Pathways, and Nutritional Implications"

_nutrients, 2025, doi:10.3390/nu17081351_

Round 1
Reviewer 1 Report
Comments and Suggestions for Authors
This is a comprehensive review of the studies of Vitamin D and colorectal cancer. However, the impression that the reader has reading some sections is that Vitamin D deficiency is causing per se CRC and not a prevention as it is nice stated in other section of the manuscript. Please revise the manuscript throughout. Moreover, the bullet’s presentation is not proper for the specific manuscript but there are proper for the tables. It would be also more easily readable if you could condensate the physiology and the pathophysiology reducing the length of the manuscript.
Author Response
Dear Reviewer,
Thank you for your valuable comments and detailed feedback. We greatly appreciate your work and suggestions, and we have made every effort to revise the manuscript in the appropriate direction. Taking into account the points you mentioned, we have revised the text, paying particular attention to the role of Vitamin D and the statements regarding CRC prevention, as well as rewriting the bullet points into text. Additionally, we have aimed to condense the sections on physiology and pathophysiology.
Please let us know if any further revisions are needed; we are happy to continue working on the manuscript.
Sincerely,
Prof. János T Varga
Reviewer 2 Report
Comments and Suggestions for Authors
Undoubtedly, the amount of material reviewed by the Authors deserves recognition. The number of studies conducted with a focus on confirming the preventive or anti-cancer properties of vitamin D in relation to CRC is enormous. However this, unfortunately, requires selective presentation of such a huge number of results, which the Authors failed to do in this article.
In the current version, the Authors "threw in" a huge amount of results, but instead of following the content designated by the detailed title, they caused chaos, creating the impression that they do not manage the content of the work at all.
The work should take into account the leading topic, specifically concerning CRC. The Authors should choose studies for presentation that include the dosing regimen, the time of vitamin D intake and/or the serum level of calcidiol (avoid the terms: “high intake” or “low intake”), define outcome measured and more precisely describe the results.
All items that the Authors include in the Tables should be discussed in the text.
The description of the results in Table 2 should be extended to include the calcidiol level. Tables 3;5 and 8 can be combined - it seems that the aim of the research, i.e. the studied relationship between vitamin D and colorectal cancers (including adenoma and polyps), is the same.
In the subsection The biological effect of vitamin D, the influence outside the musculoskeletal system should be mentioned (in a very shortened form), i.e. on the nervous, cardiovascular, respiratory, digestive, thyroid, metabolism (diabetes) systems - a whole host of target systems that the Authors have omitted. However, more emphasis should be placed in the separate subchapter, on the mechanisms specific to the action in the anticancer area, especially CRC, with a division into prophylactic or synergistic significance with chemotherapeutics. Instead of duplicating information on immunomodulatory properties, the influence on proliferation or neovascularization, i.e. general mechanisms allowing vitamin D to act in the processes of carcinogenesis or inhibiting cancer development, expand on the content contained, for example, in the fragment selectively concerning CRC (lines between 672 - 680).
The following statements appear in the work: “One of the key ways vitamin D influences CRC development is by reducing colonic inflammation. The activation of the CYP27B1 enzyme catalyzes the conversion of vitamin D into its active form, which subsequently triggers multiple anti-inflammatory pathways, mitigating the inflammatory microenvironment that contributes to CRC progression.” (337 – 340) “ or: “By improving immune responses, reducing inflammation, and inhibiting tumor cell proliferation, vitamin D supplementation enhances protection against malignancies” (lines 378 – 380) or: „Vitamin D plays a pivotal role in CRC prevention by modulating inflammatory pathways, regulating immune responses, and facilitating anti-tumor mechanisms” (lines 381 – 382). All these statements are correct, but general - the mentioned mechanisms determine the participation of vitamin D in the processes of carcinogenesis, not selectively in the case of CRC. This specificity of action is missing, although the title of the work directs its content very clearly.
The subsection on the RDA for vitamin D set in 2011 is not entirely justified if they are not commented. The subsection would be needed in the initial part of the work and in the context of presenting and commenting on the proposal that appeared regarding the update of dietary guidelines, concerning increasing the doses of vitamin D supplements to ensure preventive action in the field of chronic diseases, including cancers (Grant WB et al., 2025; Nutrients, 17, 277). However, the Authors do not address this aspect in their work.
The content of the Discussion should be moved to the main text, similarly to the subsection on adenoma and polyps in the context of the effect of vitamin D. Instead of the Discussion, Critical views and Summary should be added. In this type of work, Discussion is not necessary, because the results of other authors are presented, which simply need to be summarized. However, a subsection "Critical views (in the current version of Limitations)" is necessary, in which there should be doubts and weaknesses highlighted certainly in the works of the cited authors. This is precisely the type of discussion with the presented results.
Detailed remarks:
- Table 1 is an unnecessary form of presenting the content, and in addition, it is not very legible.
- The sentence should be reworded: (lines 61 - 63): „Vitamin D exerts both genomic effects—regulating gene expression—and rapid, non-genomic effects mediated through vitamin D receptors (VDRs) located on cell membranes” - with the current wording, it can be understood that VDR participates exclusively in non-genomic mechanisms. While it participates mainly in genomic mechanisms, bind in the promoter regions of genes with VDRE (vitamin D response elements) and thus influence transcription. Non-genomic mechanisms are directly induced by interaction with receptors, responding to the signal of thyroid hormones, estrogens or corticosteroids. In terms of detailing the mechanisms of vitamin D, the Reviewer refers to the work Żmijewski MA, 2022, Nutrients, 14, 5104.
In the summary of the review, the Reviewer suggests that the Authors organize the text rich in literature and, in terms of knowledge, very valuable text according to the following outline:
- The role of vitamin D in the body, with an indication of the role in chronic diseases.
- Current supplementary recommendations.
- Participation in the mechanisms of carcinogenesis and development of tumors (common mechanisms, such as immunoactivity, antiproliferative, anti-inflammatory effects, etc.), with an extensive part on selective CRC (mostly the current Discussion).
- Research results clearly indicating the relationship between vitamin D supplementation (dosing!) or calcidiol serum level and CRC (limit the number of Tables, avoid repetitions leading to the same conclusions in different subchapters).
- Supplementation a nd/or dietary recommendations in the light of the newly proposed (Grant WB et al., 2025, Nutrients) or resulting from the presented studies (for this purpose the doses should be included in the presented results).
- Critical views (see the commentary in the above review).
- Summary.
The reviewer hopes that the authors will make an effort to reconstruct the work, because it is valuable and enriching.
Author Response
Dear Reviewer,
We would like to express our sincere gratitude for your valuable review, assistance, and suggestions. We have prepared a comprehensive, full-scope literature summary, following the PICO criteria, which include vitamin D intake and/or serum levels. We carefully selected the original articles, as this is a thorough and comprehensive literature review study. We included all relevant research to ensure that even non-significant results are visible, providing accurate and relevant information on CRC and other precancerous conditions. Since adenomas and polyps are precancerous conditions, they are also included in the manuscript, as they can later develop into CRC.
The terms high intake and low intake are necessary to use, but we have clearly defined what constitutes high and low intake in each study, and this has been incorporated into the tables and text. Since the tables contain detailed data from the studies (a total of 50 original studies), we did not repeat the same information in the text to avoid redundancy and make the manuscript clearer.
We have expanded Table 2 with the requested levels. If we were to combine the three tables (3, 5, and 8), it would create confusion in the manuscript, as each table contains different information. Specifically: Table 3: A summary of clinical studies examining the relationship between dietary vitamin D intake and colorectal cancer risk. Table 5: A summary of studies investigating the relationship between vitamin D supplementation and colorectal cancer risk. Table 8: A summary of clinical trials examining the relationship between vitamin D and colorectal adenomas/polyps. Therefore, we decided to keep the three tables separate to maintain clarity and coherence in the manuscript.
Yes, thank you very much, we have expanded the Biological effects of vitamin D subsection to include effects beyond the musculoskeletal system, briefly touching on neurological, cardiovascular, respiratory, digestive, thyroid, and metabolic systems (e.g., diabetes).
We sincerely thank you for your valuable help. We have rephrased the general statements and improved entire chapters to better reflect the specific effects of vitamin D on CRC. The previously formulated statements were correct; however, they were too general, as the role of vitamin D in carcinogenesis is not limited to CRC. To address this, we have elaborated on how vitamin D influences CRC development, particularly through the regulation of inflammatory processes, modulation of immune responses, and anticancer mechanisms.
We have removed the subsection on the 2011 Vitamin D RDA and referenced the work by Grant WB et al., 2025; Nutrients, 17, 277, as suggested by the reviewer. Thank you very much for this suggestion.
We have removed the discussion section and replaced it with critical remarks and a summary in the manuscript.
We have incorporated the work of Żmijewski MA, 2022, Nutrients, 14, 5104, as suggested by the reviewer, to explain the mechanisms of vitamin D.
Thank you once again for your valuable review and assistance to improve the quality of our manuscript.
With kind regards,
Prof. János T Varga
Round 2
Reviewer 1 Report
Comments and Suggestions for Authors
no more comments
Author Response
Dear Reviewer,
Thank you very much!
With kind regards,
Prof. János T Varga and Co-authors from Hungary
Reviewer 2 Report
Comments and Suggestions for Authors
The revised work has gained a lot of value by directing, in fact, in accordance with the title, attention to the potential role of vitamin D in the prevention of CRC. The part concerning the mechanisms underlying the interaction of the vitamin and possible carcinogenesis in this case has been developed very thoroughly.
The Reviewer appreciates the effort of the Authors to detail the individual descriptions. The explanation of the intention of the work, presented in the letter from the Authors, was also useful. Based on the explanations, the Reviewer verified his approach to some of the reservations reported, such as the limitation of the number of Tables.
However, there were several minor inaccuracies that should be corrected or explained.
Comments:
- line 358 - 361: the indicated fragment results in a contradiction, which is actually a mental shortcut, but still requires precise expression. Vitamin D is an example of a factor that, in interaction with SIRT1, determines the effect expected in the case of CRC. However, it is necessary to emphasize here the duality of the role of SIRT1, as a tumor suppressor or as a tumor promoter. Garcia-Martinez [123] rightly refers in this respect to the works of Carafa V et al., 2019 and Ren NSX et al., 2017, which should perhaps be mentioned here.
Minor comments:
- line 334 - 335 - no literature reference;
- Table 8 - font should be adjusted.
.
Author Response
Dear Reviewer,
Thank you for acknowledging our efforts to provide detailed descriptions in the manuscript and for finding our work valuable. We are sincerely grateful for your time, your valuable contribution, and your insightful comments. We are pleased that your suggestions helped us improve specific aspects of the manuscript, and we truly appreciate your support throughout the revision process.
We have addressed the minor inaccuracies you pointed out and made the recommended corrections:
-
The fragment mentioned in lines 358–361 has been revised to avoid any potential ambiguity. We have emphasized the dual role of SIRT1 (as a tumor suppressor and a tumor promoter) and included references to the works of Carafa et al. (2019) and Ren et al. (2017), in line with the observation by Garcia-Martinez.
-
A proper literature reference has been added to lines 334–335.
-
The font size and formatting of Table 8 have been adjusted to match the rest of the tables in the manuscript.
Thank you once again for your thoughtful and constructive feedback, which has significantly contributed to the improvement of our manuscript.
Sincerely,
Prof. János T Varga and Co-authors from Hungary